# Multi-view Subspace Clustering on Topological Manifold

**Shudong Huang[1]\*, Hongjie Wu[1]\*,**
**Yazhou Ren[2], Ivor W. Tsang[3], Zenglin Xu[4], Wentao Feng[1]†, Jiancheng Lv[1]†**

[1]College of Computer Science, Sichuan University, China
[2]School of Computer Science and Engineering, UESTC, China
[3]Centre for Frontier AI Research, A\*STAR, Singapore
[4]School of Computer Science and Technology, Harbin Institute of Technology Shenzhen, China

{huangsd, Wtfeng2021, lvjiancheng}@scu.edu.cn, wuhongjie0818@gmail.com,
yazhou.ren@uestc.edu.cn, ivor.tsang@gmail.com, xuzenglin@hit.edu.cn

## Abstract

Multi-view subspace clustering aims to exploit a common affinity representation by means of self-expression. Plenty of works have been presented to boost the clustering performance, yet seldom considering the topological structure in data, which is crucial for clustering data on manifold. Orthogonal to existing works, in this paper, we argue that it is beneficial to explore the implied data manifold by learning the topological relationship between data points. Our model seamlessly integrates multiple affinity graphs into a consensus one with the topological relevance considered. Meanwhile, we manipulate the consensus graph by a connectivity constraint such that the connected components precisely indicate different clusters. Hence our model is able to directly obtain the final clustering result without reliance on any label discretization strategy as previous methods do. Experimental results on several benchmark datasets illustrate the effectiveness of the proposed model, compared to the state-of-the-art competitors over the clustering performance.

## 1 Introduction

With the evolution of multimedia, now large amounts of data are represented by multiple types (i.e., views). A document, for example, can be presented by images, audio, text. Likewise, a text can be expressed in different languages [1]. In order to integrate the complementary information among multiple views, numerous sophisticated multi-view clustering algorithms have been proposed [2, 3], which typically produce better results than single-view clustering [4, 5].

Over the past years, self-representation subspace clustering methods, which assume that each data point can be reconstructed by a linear combination of other data points, have gained broad attention. The self-representation property is able to capture the heterogeneous relationships between data and therefore produces excellent clustering results [6]. Meanwhile, self-representation property based multi-view subspace clustering algorithms have been developed up to now [7]. For instance, diversity-induced multi-view subspace clustering (DiMSC) [8] adopts the Hilbert-Schmidt independence criterion (HSIC) as a diversity term to measure the dependence of different views. By minimizing the diversity term, the subspace representations are compelled to be distinct from each other. Instead of integrating multi-view graphs into a consensus one, [9] involves the spectral clustering into their object function to derive a unified indicator matrix. Therefore, the consistent clustering results of

---

\*Equal contribution.
†Corresponding author.

36th Conference on Neural Information Processing Systems (NeurIPS 2022).

data points in different views are ensured. [10] separates the subspace representations of different views by a shared consistent representation and a set of specific representations, which is expected to better accommodate real-world data. To eliminate the redundant information of data observations or kernel matrices, [11] utilizes eigendecomposition to get the robust data representations, which helps to obtain the consensus representations of low redundancy and better clustering results. Lately, a number of anchor-based multi-view subspace clustering methods [12–14] draw significant attention because they can achieve promising performance with a large reduction in storage and computational time.

Despite the significant progress made by the aforementioned methods, there are still drawbacks that can be modified. On one hand, these methods do not consider the manifold topological structure. Considering that real-world datasets are usually sampled from a nonlinear low-dimensional manifold [15–17], it is essential to explicitly explore the topological relationship for clustering data on manifold. On the other hand, existing multi-view clustering methods usually adopt predefined similarity graphs. That is, the graph learning and subsequent multi-view clustering in these methods are separated. Thus the constructed graph may not be suitable, let alone optimal, for the subsequent clustering. It is preferred to automatically learn the similarity information between data points and involve graph learning as a part of the optimization procedure.

Regrading the deficiencies mentioned above, in this paper we argue that it is beneficial to explore the implied data manifold by learning the topological relationship between data points. To do so, we propose to seamlessly integrate multiple affinity graphs into a consensus one with the topological relevance considered. Besides, we manipulate the consensus graph by a connectivity constraint such that the connected components precisely indicate different clusters. Hence our model is able to directly obtain the final clustering result without reliance on any label discretization step. By leveraging the subtasks of affinity graph constructing, topological relevance learning, and discrete label partitioning into a unified framework, each subtask can be enhanced in a mutual reinforcement manner. An alternating iterative algorithm is carefully designed to solve the optimization problem of the proposed model. Experimental results on several benchmark datasets demonstrate the effectiveness of our method.

**Notations.** We use boldface uppercase letter, e.g., $\mathbf{M}$, to denote the matrix. $\mathbf{M}_{ij}$ represents the $ij$-th element of $\mathbf{M}$. $\left\|\cdot\right\|_F$ represents the Frobenius norm of a matrix and $\mathbf{1}$ is a column vector with all its elements equal to 1. $\mathbf{I}$ denotes the identity matrix with proper size.

## 2   Preliminary

It is well-known that real-world datasets are often sampled from a nonlinear low-dimensional manifold which is embedded in the high dimensional ambient space [15, 18–20]. Hence it is necessary to exploit the manifold structure implied within the original data.

Recently, [17] pointed out that it is preferred to boost the learning performance by making use of the manifold topological structure than the Euclidean structure. It is based on a simple yet intuitive assumption that the topological connectivities between individuals could be propagated from near to far. In other words, the spatial similarity between two individuals may be small, but their topological relevance to each other would be high if they are linked by consecutive neighbors. A visualized example is given in Figure 1. For better illustration, we visualize the topological relevance in both a 2D version (left subfigure) as well as a 3D version (right subfigure). Taking the left subfigure as an example, although the dark blue and light grey points are with low similarity in terms of spatial location and velocity, they are closely connected to each other considering the high topological relevance between them.

Instead of relying on the Euclidean structure, it is naturally expected to learn a more suitable manifold topological structure such that the intrinsic similarities can be explicitly uncovered. Considering that data points with a high similarity would share similar topological relevance, given a predefined similarity graph $\mathbf{Z} \in \mathbb{R}^{n \times n}$, where $n$ is the number of data points, [17] investigated the topological structure of data by solving

$$\min_{\mathbf{S}} \frac{1}{2} \sum_{i,j,k=1}^{n} \mathbf{Z}_{jk} \left(\mathbf{S}_{ij} - \mathbf{S}_{ik}\right)^2 + \beta \left\|\mathbf{S} - \mathbf{I}\right\|_F^2, \tag{1}$$

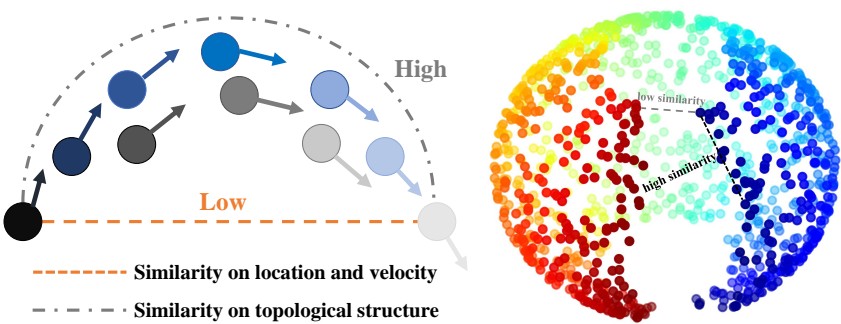

Figure 1: Illustration of topological relevance. Left: the dark blue point and the light gray point show low similarity on spatial velocity, but they keep high topological relevance to each other [17]. Right: a closer distance in the Euclidean structure does not mean higher topological relevance.

where $\beta$ is a trade-off parameter. $i$, $j$, and $k$ are data point indexes. $\mathbf{S}$ represents the target topological relationship matrix, and $\mathbf{S}_{ij}$ denotes the data point $j$'s topological relevance to $i$. Note that the first term in Eq. (1) is essentially a smoothness constraint that fits the above assumption, i.e., it guarantees the data points $j$ and $k$ share a similar topological relationship with data point $i$ if they are similar. And the second term is incorporated to prevent the trivial solution. According to Eq. (1), the topological consistency can be propagated through neighbors with high similarities, and the distant data points will keep a close relationship if they are linked by consecutive neighbors. Notwithstanding, Eq. (1) is fed with a predefined graph, which would lead to performance degradation as the predefined graph $\mathbf{Z}$ might not be optimal for subsequent learning procedures. Hence it is preferred to automatically learn similarity graphs from data.

Self-expression has been widely employed to recover the similarity graph in the form of subspace representation [6]. Given a dataset $\mathbf{X} = [\mathbf{x}_1, \mathbf{x}_2, \cdots, \mathbf{x}_n] \in \mathbb{R}^{m \times n}$ with $n$ data points and $m$ features, the self-expression based clustering problem can be defined as

$$\min_{\mathbf{Z}} \|\mathbf{X} - \mathbf{X}\mathbf{Z}\|_F^2 + \alpha \|\mathbf{Z}\|_F^2$$
$$\text{s.t. } \mathbf{Z} \geq 0, diag\left(\mathbf{Z}\right) = 0, \tag{2}$$

where $\alpha$ is a trade-off parameter and $diag\left(\mathbf{Z}\right)$ represents the vector consists of diagonal elements of $\mathbf{Z}$. It is obvious that Eq. (2) learns the neighboring data points of a data point and the corresponding weights by the sparse reconstruction from other data points. Moreover, the learned sparse representation enjoys several nice properties, e.g., the datum-adaptive ability and robustness to noise [21, 22].

Based on Eq. (2), we can easily extend it to a multi-view formulation when dealing with the multi-view data $\{\mathbf{X}^{(1)}, \mathbf{X}^{(2)}, \cdots, \mathbf{X}^{(m)}\}$, which can be formed as

$$\min_{\mathbf{Z}^{(v)}} \sum_{v=1}^{m} \left\|\mathbf{X}^{(v)} - \mathbf{X}^{(v)}\mathbf{Z}^{(v)}\right\|_F^2 + \alpha \left\|\mathbf{Z}^{(v)}\right\|_F^2$$
$$\text{s.t. } \mathbf{Z}^{(v)} \geq 0, diag\left(\mathbf{Z}^{(v)}\right) = 0, \tag{3}$$

where $m$ is the number of views, and $\mathbf{Z}^{(v)}(1 \leq v \leq m)$ denotes the affinity representation that automatically learned from the $v$-th view.

## 3 The Proposed Methodology

To explicitly incorporate the manifold topological structure into multi-view clustering, we propose to learn a topological relation graph based on Eq. (1). With the subspace representation automatically

obtained in Eq. (3), we arrive at

$$\min_{\mathbf{Z}^{(v)},\mathbf{S}} \sum_{v=1}^{m} \left\| \mathbf{X}^{(v)} - \mathbf{X}^{(v)}\mathbf{Z}^{(v)} \right\|_F^2 + \alpha \left\| \mathbf{Z}^{(v)} \right\|_F^2 + \frac{1}{2} \sum_{v=1}^{m} w_v \sum_{i,j,k=1}^{n} \mathbf{Z}_{jk}^{(v)} \left( \mathbf{S}_{ij} - \mathbf{S}_{ik} \right)^2 + \beta \left\| \mathbf{S} - \mathbf{I} \right\|_F^2$$

$$\text{s.t. } \mathbf{Z}^{(v)} \geq 0, diag\left( \mathbf{Z}^{(v)} \right) = 0, \mathbf{s}_i^T \mathbf{1} = 1, s_{ij} \geq 0,$$

(4)

where the consensus graph $\mathbf{S}$ reveals the topological relationship across multiple views and is further constrained to be non-negative and the sum of each row is 1. $w_v$ is the weight for the $v$-th views, which can be considered as a constant within each iteration and will be introduced later.

Nevertheless, the learned $\mathbf{S}$ in Eq. (4) does not contain explicit cluster structures. It is expected that the consensus graph contains exactly $c$ connected components where $c$ is the number of clusters. It sounds unrealistic that such a clear structured $\mathbf{S}$ could be achieved out of thin air. Fortunately, we can solve this problem with a useful connectivity constraint. Moreover, in Eq. (4), we see that if the $j$-th data point is connected with many similar neighbors, it will largely affect the objective value. Thus we tend to normalize the objective function so that each point can be treated equally. Considering the above concerns, we propose a normalized version of Eq. (4) with an additional connectivity constraint, which can be finally formulated as

$$\min_{\mathbf{Z}^{(v)},\mathbf{S}} \sum_{v=1}^{m} \left\| \mathbf{X}^{(v)} - \mathbf{X}^{(v)}\mathbf{Z}^{(v)} \right\|_F^2 + \alpha \left\| \mathbf{Z}^{(v)} \right\|_F^2 +$$

$$\frac{1}{2} \sum_{v=1}^{m} w_v \sum_{i,j,k=1}^{n} \mathbf{Z}_{jk}^{(v)} \left( \frac{\mathbf{S}_{ij}}{\sqrt{\mathbf{D}_{jj}^{(v)}}} - \frac{\mathbf{S}_{ik}}{\sqrt{\mathbf{D}_{kk}^{(v)}}} \right)^2 + \beta \left\| \mathbf{S} - \mathbf{I} \right\|_F^2$$

(5)

$$\text{s.t. } \mathbf{Z}^{(v)} \geq 0, diag\left( \mathbf{Z}^{(v)} \right) = 0, \mathbf{s}_i^T \mathbf{1} = 1, s_{ij} \geq 0, rank\left( \mathbf{L}_S \right) = n - c,$$

where $\mathbf{D}^{(v)}$ is the degree matrix of $\mathbf{Z}^{(v)}$, $\mathbf{L}_S$ is the Laplacian matrix of $\mathbf{S}$, and $rank\left( \mathbf{L}_S \right) = n - c$ is a rank constraint that manipulates the target graph $\mathbf{S}$ containing exactly $c$ connected components. In this way, the subsequent label discretization step can be avoided since each connected component precisely indicates an individual cluster. As for the weight $w_v$ of each view, we set it in a self-tuned way inspired by [23, 24]:

$$w_v = 1 \left/ 2\sqrt{\sum_{i,j,k=1}^{n} \mathbf{Z}_{jk}^{(v)} \left( \frac{\mathbf{S}_{ij}}{\sqrt{\mathbf{D}_{jj}^{(v)}}} - \frac{\mathbf{S}_{ik}}{\sqrt{\mathbf{D}_{kk}^{(v)}}} \right)^2}. \right.$$

(6)

It is worth mentioning the salient contributions of our model formulated in Eq. (5) as follows:

- Orthogonal to other multi-view subspace clustering approaches, our model explicitly explores the implied data manifold by learning the topological relationship across multiple views. To the best of our knowledge, this is the first work that delicately incorporates the topological structure in multi-view subspace clustering.

- Our model flawlessly integrates the subtasks including affinity graph constructing, manifold topological structure learning, and discrete label partitioning into a unified framework. Hence our model is an end-to-end single-stage learning paradigm.

- An alternating iterative algorithm is introduced to solve the optimization problem. Experiments on several benchmark datasets illustrate the effectiveness of the proposed model, compared to the state-of-the-art competitors over the clustering performance.

### 3.1 Algorithm Derivation

In order to solve the problem in Eq. (5), we derive an optimization algorithm to optimize the objective function. Since the corresponding optimization problem is not jointly convex in all variables, we choose to solve it by updating one variable while fixing other variables.

### 3.1.1 Update $\mathbf{Z}^{(v)}$ for Each View

For $\mathbf{Z}^{(v)}$, the corresponding problem is

$$\min_{\mathbf{Z}^{(v)}} \sum_{v=1}^{m} \left\|\mathbf{X}^{(v)} - \mathbf{X}^{(v)}\mathbf{Z}^{(v)}\right\|_F^2 + \alpha \left\|\mathbf{Z}^{(v)}\right\|_F^2 + \frac{1}{2} \sum_{v=1}^{m} w_v \sum_{i,j,k=1}^{n} \mathbf{Z}_{jk}^{(v)} \left(\frac{\mathbf{S}_{ij}}{\sqrt{\mathbf{D}_{jj}^{(v)}}} - \frac{\mathbf{S}_{ik}}{\sqrt{\mathbf{D}_{kk}^{(v)}}}\right)^2 \quad (7)$$

$$\text{s.t. } \mathbf{Z}^{(v)} \geq 0, diag\left(\mathbf{Z}^{(v)}\right) = 0.$$

Eq. (7) can be rewritten as

$$\min_{\mathbf{Z}^{(v)}} \sum_{v=1}^{m} \left\{ Tr\left(\mathbf{H}^{(v)}\right) + \alpha Tr\left(\mathbf{Z}^{(v)}\mathbf{Z}^{(v)T}\right) + w_v Tr\left(\mathbf{S}^T\left(\mathbf{I} - \mathbf{D}^{(v)-\frac{1}{2}}\mathbf{Z}^{(v)}\mathbf{D}^{(v)-\frac{1}{2}}\right)\mathbf{S}\right) \right\} \quad (8)$$

$$\text{s.t. } \mathbf{Z}^{(v)} \geq 0, diag\left(\mathbf{Z}^{(v)}\right) = 0,$$

where $\mathbf{H}^{(v)} = \left(\mathbf{X}^{(v)} - \mathbf{X}^{(v)}\mathbf{Z}^{(v)}\right)\left(\mathbf{X}^{(v)} - \mathbf{X}^{(v)}\mathbf{Z}^{(v)}\right)^T$. Taking the derivative of Eq. (8) w.r.t $\mathbf{Z}^{(v)}$ and setting the derivative to zero, we get following solution

$$\mathbf{Z}^{(v)} = \max\left(\left(2\mathbf{K}^{(v)} + 2\alpha\mathbf{I}\right)^{-1}\left(2\mathbf{K}^{(v)} + w_v\mathbf{G}\mathbf{G}^T\right), 0\right), \quad (9)$$

where $\mathbf{G} = \mathbf{D}^{-\frac{1}{2}}\mathbf{S}$, and $\mathbf{K}^{(v)} = \mathbf{X}^{(v)T}\mathbf{X}^{(v)}$ can be treated as a liner kernel. Note that the nonlinear kernel can also be applied in our model.

### 3.1.2 Update S

Dropping the unrelated terms of Eq. (5) w.r.t. $\mathbf{S}$, thus we need to solve

$$\min_{\mathbf{S}} \frac{1}{2} \sum_{v=1}^{m} w_v \sum_{i,j,k=1}^{n} \mathbf{Z}_{jk}^{(v)} \left(\frac{\mathbf{S}_{ij}}{\sqrt{\mathbf{D}_{jj}^{(v)}}} - \frac{\mathbf{S}_{ik}}{\sqrt{\mathbf{D}_{kk}^{(v)}}}\right)^2 + \beta \|\mathbf{S} - \mathbf{I}\|_F^2 \quad (10)$$

$$\text{s.t. } \mathbf{s}_i^T \mathbf{1} = 1, s_{ij} \geq 0, rank\left(\mathbf{L}_S\right) = n - c.$$

We see that Eq. (10) is difficult to solve due to the rank constraint. Let $\sigma_i\left(\mathbf{L}_S\right)$ be the $i$-th smallest eigenvalue of $\mathbf{L}_S$. The constraint $rank\left(\mathbf{L}_S\right) = n - c$ would be satisfied if $\sum_{i=1}^{c} \sigma_i\left(\mathbf{L}_S\right) = 0$ since $\mathbf{L}_S$ is a positive semidefinite matrix. According to Ky Fan's Theorem [25]:

$$\sum_{i=1}^{c} \sigma_i\left(\mathbf{L}_S\right) = \min_{\mathbf{F}} Tr\left(\mathbf{F}^T\mathbf{L}_S\mathbf{F}\right) \quad (11)$$

$$\text{s.t. } \mathbf{F} \in \mathbb{R}^{n \times c}, \mathbf{F}^T\mathbf{F} = \mathbf{I},$$

where $\mathbf{F} \in \mathbb{R}^{n \times c}$ denotes the cluster indicator matrix. We incorporate the constraint term $\sum_{i=1}^{k} \sigma_i\left(\mathbf{L}_S\right)$, i.e., $\min_{\mathbf{F}} Tr\left(\mathbf{F}^T\mathbf{L}_S\mathbf{F}\right)$, into the cost function, thus Eq. (10) can be rewritten as

$$\min_{\mathbf{S},\mathbf{F}} \frac{1}{2} \sum_{v=1}^{m} w_v \sum_{i,j,k=1}^{n} \mathbf{Z}_{jk}^{(v)} \left(\frac{\mathbf{S}_{ij}}{\sqrt{\mathbf{D}_{jj}^{(v)}}} - \frac{\mathbf{S}_{ik}}{\sqrt{\mathbf{D}_{kk}^{(v)}}}\right)^2 + \beta \|\mathbf{S} - \mathbf{I}\|_F^2 + 2\lambda Tr\left(\mathbf{F}^T\mathbf{L}_S\mathbf{F}\right) \quad (12)$$

$$\text{s.t. } \mathbf{s}_i^T \mathbf{1} = 1, s_{ij} \geq 0,$$

where $\lambda$ is a self-tuned parameter.

Based on Eq. (12), first we search the optimal solution of $\mathbf{F}$, which can be obtained by solving

$$\min_{\mathbf{F} \in \mathbb{R}^{n \times c}, \mathbf{F}^T\mathbf{F}=\mathbf{I}} Tr\left(\mathbf{F}^T\mathbf{L}_S\mathbf{F}\right), \quad (13)$$

which is essentially a spectral problem and $\mathbf{F}$ can be achieved by calculating the $c$ eigenvectors of $\mathbf{L}_S$ corresponding to the $c$ smallest eigenvalues.

---

**Algorithm 1:** Algorithm to solve Eq. (18)

---

**Input:** a nonzero matrix $\mathbf{A}$ and a nonzero vector $\mathbf{b}$.
Set $1 < \rho < 2$, initialize $\eta > 0$, $\mathbf{q}$.
**Output:** $\mathbf{S}$.
1: **repeat**
2:   Update $\mathbf{p}$ according to (20).
3:   Update $\mathbf{s}_i$ according to (21).
4:   Update $\eta \leftarrow \rho\eta$.
5:   Update $\mathbf{q} \leftarrow \mathbf{q} + \eta(\mathbf{s}_i - \mathbf{p})$.
6: **until** converge

---

---

**Algorithm 2:** The Algorithm for Eq. (5)

---

**Input:** Multi-view data $\{\mathbf{X}^{(1)}, \mathbf{X}^{(2)}, \ldots, \mathbf{X}^{(m)}\}$ with $m$ views, cluster number $c$, parameters $\alpha$ and $\beta$.
Initialize the weight of each view $w_v = \frac{1}{m}$.
Initialize the affinity graph $\mathbf{Z}^{(v)}$ according to Eq. (2).
Initialize the consensus graph. $\mathbf{S} = \sum_{v=1}^m w_v \mathbf{Z}^{(v)}$.
**Output:** The indicator matrix $\mathbf{S} \in \mathbb{R}^{n \times n}$ with exactly $c$ connected components.
1: **repeat**
2:   Update $\mathbf{Z}^{(v)}$ according to Eq. (9).
3:   Update $\mathbf{S}$ by Algorithm 1.
4:   Update $\mathbf{F}$ according to Eq. (13).
5:   Update $w_v$ according to Eq. (6).
6: **until** converge

---

Then we search the optimal solution of $\mathbf{S}$. It is clear that Eq. (12) w.r.t. $\mathbf{S}$ can be formulated in a vector form as

$$\min_{\mathbf{S}} \sum_{i=1}^n \left\{ \sum_{v=1}^m \frac{1}{2} w_v \sum_{j,k=1}^n \mathbf{Z}_{jk}^{(v)} \left( \frac{\mathbf{S}_{ij}}{\sqrt{\mathbf{D}_{jj}^{(v)}}} - \frac{\mathbf{S}_{ik}}{\sqrt{\mathbf{D}_{kk}^{(v)}}} \right)^2 + \beta \sum_{j=1}^n (\mathbf{S}_{ij} - \mathbf{I}_{ij})^2 + \lambda \sum_{j=1}^n \|\mathbf{f}_i - \mathbf{f}_j\|_2^2 \mathbf{S}_{ij} \right\}$$
$$\text{s.t. } \mathbf{s}_i^T \mathbf{1} = 1, s_{ij} \geq 0. \tag{14}$$

Note that Eq. (14) is independent for different $i$, thus we have the following compact formulation

$$\min_{s_{ij} \geq 0, \mathbf{s}_i^T \mathbf{1}=1} \mathbf{s}_i^T \left( \sum_{v=1}^m w_v \left( \mathbf{I} - \mathbf{D}^{(v)^{-\frac{1}{2}}} \mathbf{Z}^{(v)} \mathbf{D}^{(v)^{-\frac{1}{2}}} \right) \right) \mathbf{s}_i + \beta \|\mathbf{s}_i - \mathbf{e}_i\|_2^2 + \mathbf{s}_i^T \mathbf{u}_i, \tag{15}$$

where $\mathbf{e}_i$ denotes the $i$-th column vector of identity matrix $\mathbf{I}$, and $\mathbf{u}_i$ represents a vector with its $j$-th element $u_{ij} = \lambda \|\mathbf{f}_i - \mathbf{f}_j\|_2^2$.

Denote $\mathbf{A} = \sum_{v=1}^m w_v \left( \mathbf{I} - \mathbf{D}^{(v)^{-\frac{1}{2}}} \mathbf{Z}^{(v)} \mathbf{D}^{(v)^{-\frac{1}{2}}} \right) + \beta\mathbf{I}$ and $\mathbf{b} = 2\beta\mathbf{e}_i - \mathbf{u}_i$, Eq. (15) is equivalent to optimizing

$$\min_{s_{ij} \geq 0, \mathbf{s}_i^T \mathbf{1}=1} \mathbf{s}_i^T \mathbf{A} \mathbf{s}_i - \mathbf{s}_i^T \mathbf{b}. \tag{16}$$

We see Eq. (16) is essentially a quadratic convex optimization problem, hence it can be solved with the classical augmented Lagrangian multiplier (ALM) method [26]. Specifically, we can solve Eq. (16) by tackling its counterpart

$$\min_{s_{ij} \geq 0, \mathbf{s}_i^T \mathbf{1}=1, \mathbf{p}=\mathbf{s}_i} \mathbf{s}_i^T \mathbf{A} \mathbf{p} - \mathbf{s}_i^T \mathbf{b}. \tag{17}$$

Based on Eq. (17), the corresponding augmented Lagrangian function can be defined as

$$\min_{s_{ij} \geq 0, \mathbf{s}_i^T \mathbf{1}=1, \mathbf{p}} \mathbf{s}_i^T \mathbf{A} \mathbf{p} - \mathbf{s}_i^T \mathbf{b} + \frac{\eta}{2} \left\| \mathbf{s}_i - \mathbf{p} + \frac{1}{\eta} \mathbf{q} \right\|_2^2, \tag{18}$$

where the second term in Eq. (18) is employed to guarantee that $\mathbf{p} = \mathbf{s}_i$, $\eta$ and $\mathbf{q}$ are the penalty coefficient and parameter, respectively.

It is clear that we can solve $\mathbf{p}$ and $\mathbf{s}_i$ in an iterative updating way:

**1) Update p with fixed $\mathbf{s}_i$.** The corresponding cost function w.r.t. $\mathbf{p}$ is

$$\mathcal{L}_{\mathbf{p}} = \mathbf{s}_i^T \mathbf{A} \mathbf{p} + \frac{\eta}{2} \left\| \mathbf{s}_i - \mathbf{p} + \frac{1}{\eta} \mathbf{q} \right\|_2^2. \tag{19}$$

Setting $\frac{\partial \mathcal{L}_{\mathbf{p}}}{\partial \mathbf{p}} = 0$, we obtain

$$\mathbf{p} = \mathbf{s}_i - \frac{1}{\eta} \left( \mathbf{A}^T \mathbf{s}_i + \mathbf{q} \right). \tag{20}$$

**2) Update $\mathbf{s}_i$ with fixed p.** The corresponding cost function w.r.t. $\mathbf{s}_i$ can be reformulated as

$$\min_{s_{ij} \geq 0, \mathbf{s}_i^T \mathbf{1} = 1} \left\| \mathbf{s}_i - \mathbf{p} + \frac{1}{\eta} \mathbf{q} + \frac{\mathbf{A}\mathbf{p} - \mathbf{b}}{\eta} \right\|_2^2, \tag{21}$$

which leads to a closed-form solution and can be directly obtained by [21]. According to the ALM principles [26], $\eta$ can be exaggerated increasingly during each iteration, and $\mathbf{q}$ is updated by $\mathbf{q} \leftarrow \mathbf{q} + \eta (\mathbf{s}_i - \mathbf{p})$. Finally, we can obtain a optimal solution for $\mathbf{S}$ based on this effective ALM strategy. The detailed algorithm to solve Eq. (18) is outlined in Algorithm 2.

Up to now, with the variable $\mathbf{Z}^{(v)}$ updated by Eq. (9), $\mathbf{S}$ updated by Eq. (18), $\mathbf{F}$ updated by Eq. (13), and $w_v$ updated by Eq. (6), the overall algorithm to solve the objective in Eq. (5) can be summarized in Algorithm 2. Owing to space limitation, we give the convergence analysis and the time complexity of Algorithm 2 in the **Appendix** B and **Appendix** C, respectively.

## 4   Experiments

We evaluate the proposed method by comparing it with following state-of-the-art competitors: Diversity-induced Multi-view Subspace Clustering (**DiMSC**) [8], Auto-weighted Multiple Graph Learning (**AMGL**) [27], Multi-view Clustering with Graph Learning (**MVGL**) [28], Weighted Multi-view Spectral Clustering(**WMSC**) [29], Consistent and Specific Multi-view Subspace Clustering (**CSMSC**) [10], Graph-based Multi-view Clustering (**GMC**) [24], Large-scale Multi-View Subspace Clustering in linear time (**LMVSC**) [12], Scalable Multi-View Subspace Clustering with unified anchors (**SMVSC**) [13], and Multi-View Subspace Clustering via Co-training robust data representation (**CoMSC**) [11]. The classic graph-based algorithm, spectral clustering (SC) [18], is also included as a baseline. We apply SC by making use of the most informative view, i.e., one that achieves the best performance (denoted by $\mathrm{SC}_{best}$).

Table 1: Characteristics of all datasets. $n$, $m$, and $c$ denote the number of samples, views, and clusters, respectively. $d_v$ denotes the dimensionality of the features in the $v$-th view.

| Dataset | $n$ | $m$ | $c$ | $d_1$ | $d_2$ | $d_3$ | $d_4$ | $d_5$ | $d_6$ |
|---|---|---|---|---|---|---|---|---|---|
| 3Sources | 169 | 3 | 6 | 3560 | 3631 | 3068 | – | – | – |
| MSRC | 210 | 5 | 7 | 24 | 576 | 512 | 256 | 254 | – |
| COIL-20 | 1440 | 3 | 20 | 1024 | 3304 | 6750 | – | – | – |
| Caltech-7 | 1474 | 6 | 7 | 48 | 40 | 254 | 1984 | 512 | 928 |
| 100Leaves | 1600 | 3 | 100 | 64 | 64 | 64 | – | – | – |
| Caltech-20 | 2386 | 6 | 20 | 48 | 40 | 254 | 1984 | 512 | 928 |
| MNIST | 10000 | 3 | 10 | 30 | 9 | 30 | – | – | – |

The experiments are conducted on several benchmark datasets, namely, 3Sources, MSRC, 100Leaves, COIL-20, Caltech-7, Caltech-20, and MNIST. The detailed information of all datasets is given in **Appendix** A. The specific characteristics of these datasets are also summarized in Table 1. The parameters for comparison algorithms are set according to the recommendations in their corresponding paper. The parameter settings of our model will be introduced later. We repeat all algorithms 10 times to eliminate the effect of random factors, and report the average scores. Four widely-used metrics, i.e., clustering accuracy (ACC), Normalized Mutual Information (NMI), Purity, and F-score, are used to achieve a comprehensive evaluation.

Table 2: Clustering results of all methods on different datasets (%). The best performance is **bolded**, and the second best performance is underlined.

| Dataset | SC$_{best}$ | DiMSC | AMGL | MVGL | WMSC | CSMSC | GMC | LMVSC | SMVSC | CoMSC | **Ours** |
|---|---|---|---|---|---|---|---|---|---|---|---|
| ACC | | | | | | | | | | | |
| 3Sources | 53.67 | 76.33 | 44.14 | 42.54 | 57.75 | 78.34 | 69.23 | 50.18 | 43.14 | 64.26 | **81.07** |
| MSRC | 58.95 | 72.38 | 70.67 | 70.48 | 69.00 | 80.48 | 74.76 | 74.71 | 81.43 | 80.86 | **85.24** |
| COIL-20 | 72.75 | 76.15 | 79.30 | 75.21 | 76.58 | 75.06 | 79.10 | 75.56 | 61.07 | 71.90 | **80.42** |
| Caltech-7 | 48.58 | 41.51 | 64.66 | 56.38 | 38.95 | 62.08 | 69.20 | 60.91 | 57.22 | 64.65 | **77.61** |
| 100Leaves | 69.62 | 47.87 | 79.09 | 54.12 | 78.23 | 76.78 | 82.38 | 67.32 | 38.03 | 78.75 | **83.56** |
| Caltech-20 | 41.74 | 28.45 | 49.69 | 57.29 | 33.98 | 47.47 | 45.64 | 47.10 | 61.36 | 53.32 | **68.99** |
| MNIST | 52.74 | 51.79 | 85.10 | 30.55 | 51.91 | 50.64 | 84.37 | 71.45 | 77.16 | 69.65 | **87.44** |
| NMI | | | | | | | | | | | |
| 3Sources | 49.99 | 63.77 | 18.35 | 27.11 | 49.33 | 70.75 | 54.80 | 30.51 | 24.21 | 59.32 | **70.81** |
| MSRC | 46.81 | 60.08 | 66.80 | 58.18 | 59.53 | 71.43 | 74.21 | 65.55 | 70.18 | 74.08 | **77.35** |
| COIL-20 | 81.91 | 83.02 | 91.43 | 83.80 | 84.16 | 84.17 | 91.79 | 83.24 | 73.06 | 81.42 | **91.90** |
| Caltech-7 | 28.99 | 32.10 | 52.76 | 51.63 | 28.08 | 51.82 | 60.56 | 44.33 | 44.96 | 55.96 | **64.51** |
| 100Leaves | 86.17 | 70.98 | 90.48 | 63.96 | 90.44 | 89.05 | 90.25 | 84.64 | 64.92 | 90.42 | **92.48** |
| Caltech-20 | 45.47 | 27.59 | 54.47 | 58.59 | 41.81 | 57.83 | 38.46 | 49.21 | 57.56 | **59.38** | 56.53 |
| MNIST | 47.13 | 34.08 | 76.08 | 24.04 | 47.31 | 46.13 | 76.39 | 63.46 | 62.40 | 64.80 | **77.49** |
| Purity | | | | | | | | | | | |
| 3Sources | 71.18 | 80.47 | 49.94 | 48.46 | 71.48 | 83.67 | 74.56 | 75.74 | 53.08 | 72.01 | **84.62** |
| MSRC | 60.00 | 72.38 | 74.14 | 70.48 | 71.38 | 80.48 | 79.05 | 75.33 | 81.43 | 81.76 | **85.24** |
| COIL-20 | 75.25 | 78.94 | 84.37 | 77.78 | 78.19 | 77.56 | 84.79 | 79.08 | 61.72 | 78.94 | **85.00** |
| Caltech-7 | 79.61 | 76.11 | 84.83 | 86.84 | 79.58 | 86.95 | 88.47 | 70.98 | 85.80 | 72.73 | **88.60** |
| 100Leaves | 72.94 | 50.47 | 83.42 | 57.44 | 80.55 | 79.44 | 85.06 | 77.39 | 39.49 | 85.44 | **86.01** |
| Caltech-20 | 70.54 | 54.83 | 68.33 | 74.85 | 67.29 | **77.91** | 55.49 | 52.48 | 71.32 | 61.12 | 75.02 |
| MNIST | 56.27 | 52.37 | 85.43 | 30.55 | 55.94 | 54.22 | 84.37 | 77.00 | 77.16 | 76.38 | **87.44** |
| F-score | | | | | | | | | | | |
| 3Sources | 48.49 | 70.68 | 38.18 | 44.75 | 50.79 | 73.17 | 60.47 | 41.87 | 38.46 | 60.49 | **75.25** |
| MSRC | 43.88 | 58.61 | 62.22 | 54.56 | 57.52 | 70.13 | 69.68 | 64.71 | 69.36 | 71.35 | **75.29** |
| COIL-20 | 69.09 | 72.27 | 75.95 | 71.43 | 73.40 | 70.75 | 79.42 | 70.23 | 53.68 | 66.33 | **82.29** |
| Caltech-7 | 40.01 | 42.26 | 61.41 | 59.77 | 37.78 | 61.74 | 72.17 | 56.37 | 55.46 | 64.92 | **79.77** |
| 100Leaves | 61.94 | 33.12 | 59.14 | 8.58 | 72.63 | 69.64 | 50.42 | 58.20 | 23.20 | **73.19** | 69.29 |
| Caltech-20 | 33.21 | 20.10 | 39.78 | 47.05 | 30.54 | 42.30 | 34.03 | 39.78 | **66.27** | 47.72 | 53.13 |
| MNIST | 41.53 | 32.80 | 74.99 | 24.46 | 41.08 | 41.41 | 74.43 | 59.42 | 62.39 | 61.04 | **77.67** |

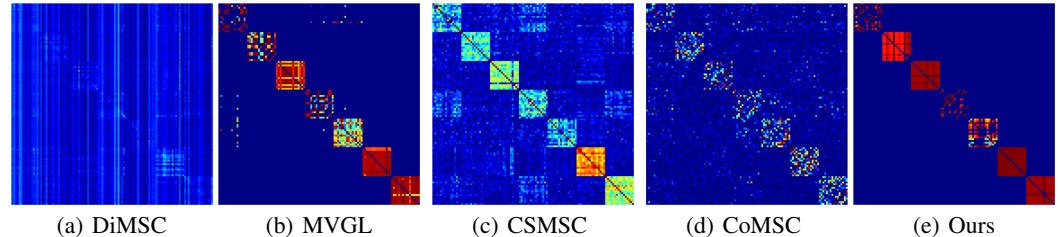

| (a) DiMSC | (b) MVGL | (c) CSMSC | (d) CoMSC | (e) Ours |
|---|---|---|---|---|

Figure 2: Target consensus graph of dataset 100Leaves (first 7 categories) learned by different methods.

## 4.1 Results and Analysis

The clustering performance of all methods on four metrics is shown in Table 2. As we see, our proposed method achieves the best performance in the majority of cases, which validates the effectiveness of our method. Moreover, it is obvious that the improvement is remarkable. On dataset MSRC, for instance, our method even achieves substantial improvements around 4.7%, 4.2%, 4.3%, and 10.5% over the most competitive methods in terms of ACC, NMI, Purity, and F-score, respectively. Note that DiMSC, CSMSC, LMVSC, SMVSC, and CoMSC are all multi-view subspace clustering methods that adopt the Euclidean structure. From the results, it is obvious our method that employs manifold topological structure is much better than that in terms of clustering performance, which verifies our assumption that manifold topological structure is more suitable to explicitly uncover the intrinsic similarities. Taking the dataset 100Leaves as an example, we visualize the target graph learned by different methods. For better visualization, we plot the first seven categories of dataset 100Leaves in Figure 2. As can be seen, DiMSC cannot even find the block diagonal structure of the target graph. MVGL, CSMSC, and CoMSC are able to search the block diagonal structure but contain lots of noise obviously. On the contrary, our model almost achieves a pure structured graph with a much clear clustering structure.

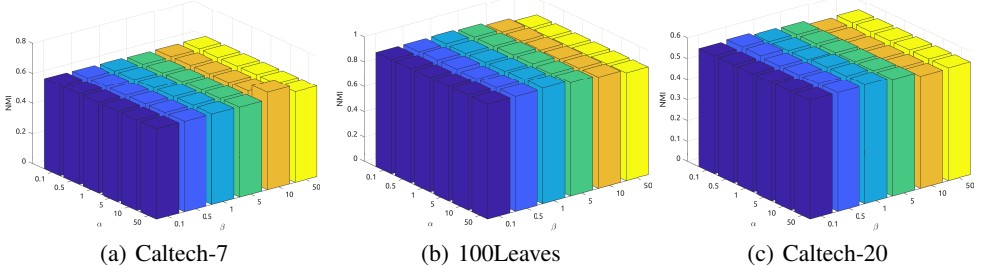

(a) Caltech-7          (b) 100Leaves          (c) Caltech-20

Figure 3: NMI w.r.t. $\alpha$ and $\beta$.

## 4.2 Parameter Analysis

This section investigates the sensitivity of our method with respect to different parameter settings. Note that $\lambda$ involved in Eq. (12) is a self-tuned parameter, which can be set in a heuristic way. That is, we can initialize $\lambda$ to a random positive value (e.g., $\lambda = 10$), if the connected components of $\mathbf{S}$ is greater than $c$ in current iteration, we set $\lambda \leftarrow \frac{\lambda}{2}$, and if it is less than $c$, we set $\lambda \leftarrow 2\lambda$. This strategy guarantees $\mathbf{S}$ contains a clear clustering structure with exact $c$ connected components. Hence we only need to set the parameters $\alpha$ and $\beta$ properly. Here we empirically search them in the range [0.1,0.5,1,5,10,50] for simplicity. The clustering performance of three datasets is shown in Figure 3 (owing to space limitation, the results of other datasets are attached in **Appendix** D), we can find that the clustering results of our method are relatively stable for different parameter settings, which demonstrates the robustness of our model. As introduced before, the target consensus graph $\mathbf{S}$ can be treated as an indicator matrix, where the points from the same cluster are connected to the same component. Once we obtain $\mathbf{S}$, the cluster label of each data point can be directly assigned without any postprocessing. Hence our method is very stable. In general, we could obtain a promising clustering performance by setting $\alpha = \beta = 10$ in practical applications.

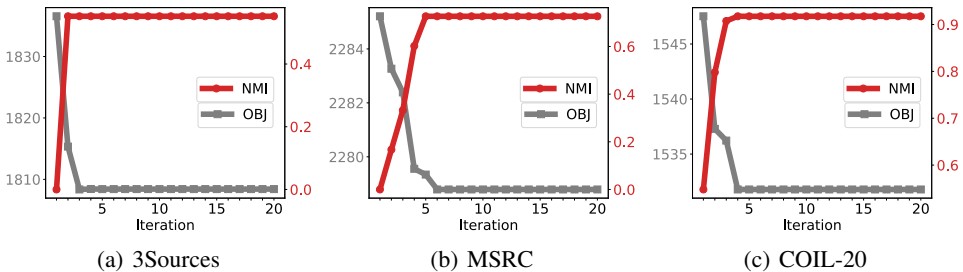

(a) 3Sources          (b) MSRC          (c) COIL-20

Figure 4: Convergence speed of our method, where OBJ denotes the objective value.

## 4.3 Convergence Study

Here we verify the convergence property of the proposed algorithm. The convergence curves along with the corresponding performance of our algorithm on 3Sources, MSRC, and COIL-20 datasets are recorded in Figure 4 (the convergence curves of other datasets are attached in **Appendix** D), where the red line denotes the NMI of our method and the grey line indicates the objective value. As we can see, the proposed optimization algorithm is efficient and converges very fast, usually within 10 iterations, which empirically illustrates the efficiency of the proposed algorithm.

## 5 Conclusion

In this paper, we propose to explore the implied data manifold by learning the topological relationship between data points. To do so, we seamlessly integrate multiple affinity graphs into a consensus one with the topological relevance considered. Besides, we manipulate the consensus graph by

a connectivity constraint such that the connected components precisely indicate different clusters. Hence our model can directly obtain the discrete result without any postprocessing. An alternating iterative algorithm is carefully designed to solve the optimization problem of the proposed model. Experimental results have shown that (1) manifold topological structure is suitable to explicitly uncover the intrinsic similarities, thus beneficial for multi-view subspace clustering tasks; (2) our model is quite robust with respect to different parameter settings, which demonstrates its stability; (3) the proposed optimization algorithm is very efficient and converges fast. However, note that our model cannot deal with the nonlinear data, which can be considered in future works. As mentioned in Subsection 3.1.1, the kernel strategy would be an option to explore nonlinear structure. Furthermore, we are also interested in extending the proposed framework to other machine learning frameworks, such as semi-supervised learning and deep learning.

## Acknowledgments

We thank the anonymous reviewers for their helpful comments and suggestions. This work is supported by the Key Program of National Science Foundation of China (Grant No. 61836006), partially supported by the National Science Foundation of China under Grant 62106164 and 62106161, the 111 Project under Grant B21044, and the Sichuan Science and Technology Program under Grants 2021ZDZX0011 and 2022YFG0188.

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
