# Appendix:

## A  Datasets

Several benchmark multi-view datasets are adopted in our experiments. **3Sources**[3] is collected from three news sources, i.e., Reuters, BBC, and The Guardian. There are 948 news articles covering 416 different news stories. Among them, 169 news were reported in all three sources and each news was annotated with one of six topical labels: business, health, politics, entertainment, sport, and technology. **MSRC** is comprised of 240 images in eight classes. We select seven classes with each class containing 30 images. For each image, five visual features are extracted for a comprehensive description. Caltech101 is an object recognition dataset with 101 categories. This dataset is represented by 6 types of features. Following [23], we select 1474 images within 7 classes (**Caltech-7**) and 2386 images within 20 classes (**Caltech-20**). **100Leaves**[4] is the one-hundred plant leaves data set. It contains 1600 data samples and each of which is described by fine-scale margin, shape descriptor, and texture histogram. **COIL-20** is a subset of a object database[5] includes 100 categories. Images of each object were taken 5-degree apart as the object is rotated on a turntable and each object has 72 gray images. Each image is described by three types of features. **MNIST** is a handwritten digits database[6] that is represented by three views. We select 10000 data samples includes 10 categories in this paper, and there are 1000 data samples per category.

The specific characteristics of the datasets are given in Table 1.

## B  Convergence Analysis

In this subsection, we prove the convergence of problem (10) and Alg. 2. We will show that our algorithm can find a local optimal solution. Before we prove its convergence, first we introduce an important lemma as follows [30]:

**Lemma 1** *For any positive real number $q$ and $t$, the following inequality holds:*

$$\sqrt{q} - \frac{q}{2\sqrt{t}} \leq \sqrt{t} - \frac{t}{2\sqrt{t}}.$$

**Proof 1** *It is obvious that inequality $(\sqrt{q} - \sqrt{t})^2 \geq 0$, thus we have*
$(\sqrt{q} - \sqrt{t})^2 \geq 0 \Rightarrow q - 2\sqrt{qt} + t \geq 0 \Rightarrow \sqrt{q} - \frac{q}{2\sqrt{t}} \leq \frac{\sqrt{t}}{2} \Rightarrow \sqrt{q} - \frac{q}{2\sqrt{t}} \leq \sqrt{t} - \frac{t}{2\sqrt{t}}$
*which completes the proof.*

**Theorem 1** *In each iteration, the updated* $\mathbf{S}$ *will monotonically decrease the objective in Eq. (10), which generally makes the solution converge to the local optimum of Eq. (10).*

**Proof 2** *Suppose the alternatively updated* $\mathbf{S}$ *is* $\widetilde{\mathbf{S}}$ *in each iteration. By solving Eq. (18), we get*

$$\widetilde{\mathbf{S}} = \arg\min_{\mathbf{S}} \left\{ \frac{1}{2} \sum_{v=1}^{m} w_v \sum_{i,j,k=1}^{n} \mathbf{Z}_{jk}^{(v)} \left( \frac{\mathbf{S}_{ij}}{\sqrt{\mathbf{D}_{jj}^{(v)}}} - \frac{\mathbf{S}_{ik}}{\sqrt{\mathbf{D}_{kk}^{(v)}}} \right)^2 + \beta \left\| \mathbf{S} - \mathbf{I} \right\|_F^2 \right\}$$

---

[3]http://mlg.ucd.ie/datasets/3sources.html
[4]https://archive.ics.uci.edu/ml/datasets/One-hundred+plant+ species+leaves+data+set
[5]http://www.cs.columbia.edu/CAVE/software/softlib/coil-100.php
[6]http://yann.lecun.com/exdb/mnist/

*According to Eq. (6), i.e.,* $w_v = 1 \Big/ 2\sqrt{\sum\limits_{i,j,k=1}^{n} \mathbf{Z}_{jk}^{(v)} \left( \frac{\mathbf{S}_{ij}}{\sqrt{\mathbf{D}_{jj}^{(v)}}} - \frac{\mathbf{S}_{ik}}{\sqrt{\mathbf{D}_{kk}^{(v)}}} \right)^2}$ *, we get*

$$\frac{1}{2} \sum_{v=1}^{m} \frac{\sum\limits_{i,j,k=1}^{n} \mathbf{Z}_{jk}^{(v)} \left( \frac{\widetilde{\mathbf{S}}_{ij}}{\sqrt{\mathbf{D}_{jj}^{(v)}}} - \frac{\widetilde{\mathbf{S}}_{ik}}{\sqrt{\mathbf{D}_{kk}^{(v)}}} \right)^2}{2\sqrt{\sum\limits_{i,j,k=1}^{n} \mathbf{Z}_{jk}^{(v)} \left( \frac{\mathbf{S}_{ij}}{\sqrt{\mathbf{D}_{jj}^{(v)}}} - \frac{\mathbf{S}_{ik}}{\sqrt{\mathbf{D}_{kk}^{(v)}}} \right)^2}} + \mathcal{G}\left( \widetilde{\mathbf{S}} \right)$$

$$\leq \frac{1}{2} \sum_{v=1}^{m} \frac{\sum\limits_{i,j,k=1}^{n} \mathbf{Z}_{jk}^{(v)} \left( \frac{\mathbf{S}_{ij}}{\sqrt{\mathbf{D}_{jj}^{(v)}}} - \frac{\mathbf{S}_{ik}}{\sqrt{\mathbf{D}_{kk}^{(v)}}} \right)^2}{2\sqrt{\sum\limits_{i,j,k=1}^{n} \mathbf{Z}_{jk}^{(v)} \left( \frac{\mathbf{S}_{ij}}{\sqrt{\mathbf{D}_{jj}^{(v)}}} - \frac{\mathbf{S}_{ik}}{\sqrt{\mathbf{D}_{kk}^{(v)}}} \right)^2}} + \mathcal{G}\left( \mathbf{S} \right),$$

*where* $\mathcal{G}\left( \mathbf{S} \right) = \beta \left\| \mathbf{S} - \mathbf{I} \right\|_F^2$.

*Based on Lemma 1, we have*

$$\frac{1}{2} \sum_{v=1}^{m} \sqrt{\sum\limits_{i,j,k=1}^{n} \mathbf{Z}_{jk}^{(v)} \left( \frac{\widetilde{\mathbf{S}}_{ij}}{\sqrt{\mathbf{D}_{jj}^{(v)}}} - \frac{\widetilde{\mathbf{S}}_{ik}}{\sqrt{\mathbf{D}_{kk}^{(v)}}} \right)^2} - \frac{1}{2} \sum_{v=1}^{m} \frac{\sum\limits_{i,j,k=1}^{n} \mathbf{Z}_{jk}^{(v)} \left( \frac{\widetilde{\mathbf{S}}_{ij}}{\sqrt{\mathbf{D}_{jj}^{(v)}}} - \frac{\widetilde{\mathbf{S}}_{ik}}{\sqrt{\mathbf{D}_{kk}^{(v)}}} \right)^2}{2\sqrt{\sum\limits_{i,j,k=1}^{n} \mathbf{Z}_{jk}^{(v)} \left( \frac{\mathbf{S}_{ij}}{\sqrt{\mathbf{D}_{jj}^{(v)}}} - \frac{\mathbf{S}_{ik}}{\sqrt{\mathbf{D}_{kk}^{(v)}}} \right)^2}}$$

$$\leq \frac{1}{2} \sum_{v=1}^{m} \sqrt{\sum\limits_{i,j,k=1}^{n} \mathbf{Z}_{jk}^{(v)} \left( \frac{\mathbf{S}_{ij}}{\sqrt{\mathbf{D}_{jj}^{(v)}}} - \frac{\mathbf{S}_{ik}}{\sqrt{\mathbf{D}_{kk}^{(v)}}} \right)^2} - \frac{1}{2} \sum_{v=1}^{m} \frac{\sum\limits_{i,j,k=1}^{n} \mathbf{Z}_{jk}^{(v)} \left( \frac{\mathbf{S}_{ij}}{\sqrt{\mathbf{D}_{jj}^{(v)}}} - \frac{\mathbf{S}_{ik}}{\sqrt{\mathbf{D}_{kk}^{(v)}}} \right)^2}{2\sqrt{\sum\limits_{i,j,k=1}^{n} \mathbf{Z}_{jk}^{(v)} \left( \frac{\mathbf{S}_{ij}}{\sqrt{\mathbf{D}_{jj}^{(v)}}} - \frac{\mathbf{S}_{ik}}{\sqrt{\mathbf{D}_{kk}^{(v)}}} \right)^2}}.$$

*By summing over the above two equations in the two sides, we obtain*

$$\frac{1}{2} \sum_{v=1}^{m} \sqrt{\sum\limits_{i,j,k=1}^{n} \mathbf{Z}_{jk}^{(v)} \left( \frac{\widetilde{\mathbf{S}}_{ij}}{\sqrt{\mathbf{D}_{jj}^{(v)}}} - \frac{\widetilde{\mathbf{S}}_{ik}}{\sqrt{\mathbf{D}_{kk}^{(v)}}} \right)^2} + \mathcal{G}\left( \widetilde{\mathbf{S}} \right)$$

$$\leq \frac{1}{2} \sum_{v=1}^{m} \sqrt{\sum\limits_{i,j,k=1}^{n} \mathbf{Z}_{jk}^{(v)} \left( \frac{\mathbf{S}_{ij}}{\sqrt{\mathbf{D}_{jj}^{(v)}}} - \frac{\mathbf{S}_{ik}}{\sqrt{\mathbf{D}_{kk}^{(v)}}} \right)^2} + \mathcal{G}\left( \mathbf{S} \right),$$

*which completes the prove. That is, the objective function value will monotonically decreases in each iteration of updating* $\mathbf{S}$.

In Alg. 2, we can obtain the closed-form solutions with respect to $\mathbf{Z}^{(v)}$, $w_v$, and $\mathbf{F}$, as described in the main paper. ALM optimization theory [31] guarantees that the iterations will make the optimization process converge. In a word, the updating of all variables with iterative optimization steps will monotonically decrease toward the lower bound of the objective function in (5).

## C   Time complexity Analysis

As shown in Alg. 2, there are four steps that mainly determine the complexity of our model. Recall that $n$, $m$ and $c$ denote the number of data points, views, and clusters, respectively. We summarize the computational complexity of each step in Table 3.

In practical, we have $m \ll n$ and $c \ll n$, thus the overall complexity is $\mathcal{O}\left( n^2 \right)$, which is in the same line with the classical graph-based methods and hence is acceptable.

| Steps | Calculation | Complexity |
|---|---|---|
| Eq. (9) | update $\mathbf{Z}^{(v)}$ | $\mathcal{O}\left(n^2mc\right)$ |
| ALM | update $\mathbf{S}$ | $\mathcal{O}\left(nm^2c+nmc\right)$ |
| Eq. (13) | $c$ eigenvectors of $\mathbf{L}_S$ | $\mathcal{O}\left(n^2c\right)$ |
| Eq. (6) | view weight $w_v$ | $\mathcal{O}\left(n^2mc\right)$ |
| Total | $\approx \mathcal{O}\left(n^2\right)$ | |

Table 3: Details of computational complexity.

# D   Additional Experiments

## D.1   Computational performance

Given the computational complexity of our algorithm theoretical analyzed above, here we empirically compare the computational speed of our method with other multi-view subspace clustering approaches. The execution time of all algorithms on a machine with 2.60GHz Intel Xeon Gold 6240 CPU and 256GB RAM is shown in Table 4. We see that LMVSC and SMVSC are the two timesaving-most algorithms, especially on large dataset MNIST. On the other hand, DiMSC, CoMSC, and MVGL are the three time-consuming-most algorithms, which is slower than SMVSC by nearly two orders of magnitude. Generally speaking, our algorithm is faster than DiMSC, CoMSC, and MVGL, slower than LMVSC and SMVSC but in line with other methods, which validates the efficiency of the proposed algorithm.

| Datasets | DiMSC | AMGL | MVGL | WMSC | CSMSC | GMC | LMVSC | SMVSC | CoMSC | Ours |
|---|---|---|---|---|---|---|---|---|---|---|
| 3Sources | 0.97 | 0.11 | 0.31 | 0.17 | 0.28 | 0.20 | 6.50 | 3.61 | 2.83 | 0.14 |
| MSRC | 1.94 | 0.17 | 0.55 | 0.20 | 0.20 | 0.28 | 7.12 | 2.21 | 4.84 | 0.22 |
| COIL-20 | 51.05 | 6.49 | 11.56 | 3.03 | 6.14 | 5.31 | 29.59 | 29.84 | 25.44 | 5.68 |
| Caltech-7 | 165.44 | 9.39 | 18.34 | 7.75 | 5.76 | 6.49 | 19.72 | 9.81 | 39.69 | 10.35 |
| 100Leaves | 58.98 | 12.65 | 13.94 | 4.20 | 5.51 | 3.50 | 11.94 | 12.34 | 78.82 | 5.98 |
| Caltech-20 | 460.37 | 26.32 | 59.02 | 20.63 | 16.16 | 16.30 | 32.31 | 17.68 | 206.16 | 31.76 |
| MNIST | 46834.95 | 5052.35 | 12348.70 | 4151.89 | 2830.16 | 2262.57 | 572.81 | 77.69 | 38493.93 | 4136.35 |

Table 4: Comparison of computational performance on all datasets (seconds).

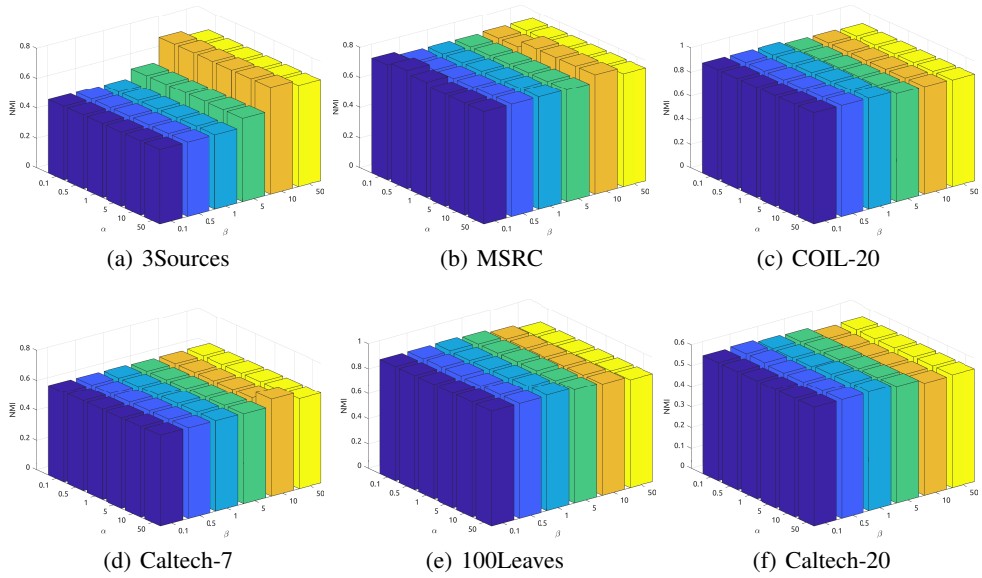

(a) 3Sources        (b) MSRC        (c) COIL-20

(d) Caltech-7        (e) 100Leaves        (f) Caltech-20

Figure 5: NMI w.r.t. $\alpha$ and $\beta$ on different datasets.

## D.2 Parameter Analysis

This section investigates the influence of different parameter settings to clustering performance. As mentioned in the main paper, $\lambda$ involved in Eq. (12) is a self-tuned parameter, and can be set in a heuristic way. That is, we can initialize $\lambda$ to a random positive value (e.g., $\lambda = 10$), then we automatically halve (i.e., $\lambda = \lambda/2$) or double (i.e., $\lambda = \lambda * 2$) it when the number of connected components is greater or smaller than the cluster number $c$ in each iteration. As a result, this strategy guarantees that the obtained graph contains a clear clustering structure. As a result, we only need to set the parameters $\alpha$ and $\beta$ properly. We empirically search both $\alpha$ and $\beta$ in the range [0.1,0.5,1,5,10,50] for simplicity. As shown in Figure 5, we can find that the clustering results of our method are relatively stable with respect to different parameter settings, which demonstrates the robustness of our model. In general, we could obtain a promising clustering performance by setting $\alpha = \beta = 10$ in practical applications.

## D.3 Convergence Study

Besides, as shown in the main paper, our empirical experiments also validate the convergence of the proposed method. The convergence curves along with the corresponding performance of our algorithm on different datasets are recorded in Figure 6, where the red line denotes the NMI of our method and the grey line indicates the objective value. As we can see, the proposed optimization algorithm is efficient and converges very fast, usually within 10 iterations, which empirically illustrates the efficiency of the proposed algorithm.

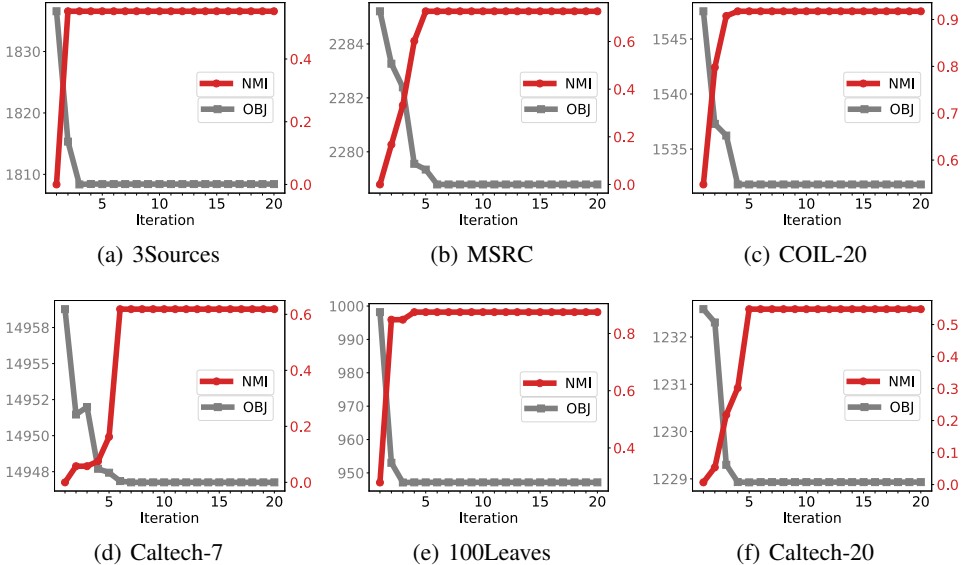

Figure 6: Convergence analysis of the proposed method, where OBJ denotes the objective value.

## D.4 Clustering Results Visualization

In this section, we visualize the consensus graph learned by the proposed method of all the datasets with t-SNE [32] in Figure 7. As can be seen, the learned consensus graph generally contains a compact and accurate class cluster, which again showcases the effectiveness of our method by considering the manifold topological structure.

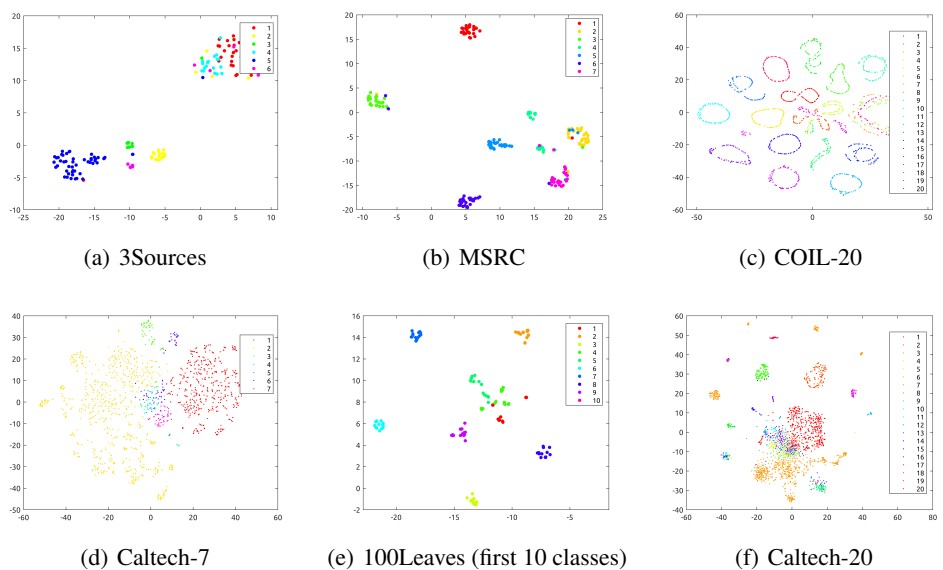

(a) 3Sources      (b) MSRC      (c) COIL-20

(d) Caltech-7      (e) 100Leaves (first 10 classes)      (f) Caltech-20

Figure 7: Visualization of the clustering results with t-SNE.