# OpenReview forum: "Multi-view Subspace Clustering on Topological Manifold"
_NeurIPS.cc/2022/Conference — NeurIPS 2022 Accept_

### Official Review · Reviewer_1etV · 2022-07-03

**Rating:** 5
**Confidence:** 5
**Soundness:** 3 good
**Presentation:** 3 good
**Contribution:** 3 good

**Summary:**

To explore the benefits of the topological structure in multi-view data, this paper is to integrate multiple affinity graphs into a consensus one with the topological manifold for multi-view subspace clustering. Experimental results on several benchmark datasets illustrate the effectiveness of the proposed model.

**Questions:**

Please see the weaknesses.

**Limitations:**

The novelty of the proposed method seems limited

**Strengths And Weaknesses:**

Strengths:
1. This paper is well-written and is easy to follow.
2. This paper proposes a multi-view subspace clustering method based on topological manifold.
3. The proposed method achieves good performance compared to the state-of-the-art competitors over the clustering performance.

Weakness:
1. The novelty of the proposed method seems limited since it is simple to combine the multi-view learning formulation in Eq. (4) with the topological structure in Eq. (3). Moreover, Eq. (4) is popular in multi-view subspace clustering and Eq. (3) has been proposed in [17]. Which part is introduced in this paper?
2. In fact, Fig. 1 is same to the Fig. 2 of [17]. So, please cite this paper in the caption of Fig. 1.
3. Experiment on the topological manifold is missing. I suggest one experiment to show the topological manifold.

---

> ### Author Response · Authors · 2022-08-02
> **Response to Reviewer 1etV**
>
> Thanks for your valuable comments.
>
> R1: During our investigation, we observe that the intrinsic connections of data points can be better learned with their topological relevance considered. So, we explore the implied data manifold by learning the topological relationship between data points. Although some of the techniques have been proposed, there are innovations and contributions compared with existing works. We summarize the distinct aspects of our method as follows:
>
> Motivation: considering that the topological relevance of two points could be propagated from near to far through consecutive neighbors, we propose to learn the topological structure for better clustering performance.
>
> Model: we propose an effective model to learn the topological structure, and innovatively introduce the model to the multi-view domain. Besides, considering that if a data point is connected with many similar neighbors, it will largely affect the objective value. Hence we further upgrade the model to a normalized version such that each point can be treated equally. In general, we learn the topological structure of each view and integrate them into a structured consensus graph with a carefully-designed objective function. Our model accomplishes the subtasks including affinity graph constructing, manifold topological structure learning, and discrete label partitioning into a unified framework. With the original input multi-view data, our model directly outputs the discrete cluster indicator matrix (in the form of the structured consensus graph). Hence it is an end-to-end single-stage learning model.
>
> Algorithm: an effective algorithm is meticulously designed for optimization, where each subproblem could be settled with an optimal solution in an iterative way.
>
> Results: we present extensive experimental results on several benchmark multi-view datasets including clustering performance comparison, visualization of consensus graph and topological manifold, parameter analysis, and convergence analysis. All the results certificate the effectiveness and superiority of our method.
>
> R2: Thanks for the suggestion, we've cited Ref.[17] in the caption in the reversion. The additional contents of the caption of Fig. 1 are as follows: "Illustration of topological relevance. Left: the dark blue point and the light gray point show low similarity on spatial velocity, but they keep high topological relevance to each other Ref.[17]. Right: a closer distance in the Euclidean structure does not mean higher topological relevance." Note that Fig. 1 in the paper aims to illustrate that the manifold topological structure is better to measure the similarity between the data than the Euclidean structure. Our Fig. 1 is inspired by the Fig. 2 of Ref.[17], but not exactly in line with it. In our figure, we represent their topological relationship by the variations in color. Besides, we've further added a new 3D subfigure in Fig.1 in the revised version, in which we aim to stereoscopically illustrate the topological relevance between data points.
>
> R3: We've presented the experiment to visualize the data manifold of the proposed method in the revised version (Appendix D.4).
> As shown in Fig. 7, it is clear that the real-world datasets naturally exhibit a manifold topological structure. Our empirical studies also corroborate our assumption, and demonstrate that it is beneficial to explore the topological structure for better clustering performance. In order to further provide an intuitive illustration of the effectiveness of our model, we apply t-distributed Stochastic Neighbor Embedding (t-SNE) to show the distribution of the target consensus graph of our model. As can be seen in Fig. 8, the learned consensus graph generally contains a compact and accurate class cluster, which again showcases the effectiveness of our method by taking the manifold topological structure into consideration.

---

> ### Author Response · Authors · 2022-08-10
> **To Reviewer 1etV**
>
> Dear Reviewer 1etV:
>
> Thanks a lot for reviewing our paper and giving us valuable comments.
>
> We have tried our best to answer all the questions according to the comments. We sincerely hope that our responses can address all your concerns. Is there anything that needs us to further clarify the given concerns?
>
> Thanks again for your great efforts.

---

### Official Review · Reviewer_wnN6 · 2022-07-06

**Rating:** 7
**Confidence:** 5
**Soundness:** 4 excellent
**Presentation:** 3 good
**Contribution:** 2 fair

**Summary:**

To explore the implied data manifold by learning the topological relationship between data points, this paper proposes to integrate multiple affinity graphs into a consensus one with the topological relevance considered. And to get the discrete result directly, a connectivity constraint is included in the proposed model. Detailed analyses about the optimization and convergence are provided. Solid experimental results are shown to demonstrate the efficiency and stability of the proposed algorithm.

**Questions:**


1. Is there any assumption of the algorithm?
2. Why there is no synthetic dataset with topological clusters used to verify the performance of the proposed algorithm.
3. How to determine the range of parameters \alpha and \beta?
4. What are linear data and nonlinear data?

**Limitations:**

Yes, the authors have addressed the limitations.

**Strengths And Weaknesses:**

Strengths:
1.	This paper is well organized and the motivation is clear to understand.
2.	The proposed algorithm, the optimization procedure, and the corresponding analysis are convinced.
3.	The empirical evaluation shows significantly better results than existing related clustering algorithms.
Weaknesses:
1.	It could be better to analyze the properties and behaviors of the proposed algorithms.
2.	Introduction section, the motivation for the proposed algorithm is ambiguous.
3.	There are some writing problems in the current manuscript.

---

> ### Author Response · Authors · 2022-08-02
> **Response to Reviewer wnN6**
>
> Thanks for your valuable comments.
>
> R1: Yes, there are some assumptions in our paper. On the one hand, the motivation of this paper is based on an intuitive assumption that real-world datasets are often sampled from a nonlinear low-dimensional manifold [1-4] and the topological connectivities between individuals could be propagated from near to far. On the other hand, we learn the similarity graph $\mathbf{Z}$ by the self-representation property, which assumes that each sample can be represented as a linear combination of other samples, which is a widely-used assumption in subspace learning.
>
> R2: We did not include any synthetic dataset for the following reasons. First, our empirical studies corroborate our theoretical findings, and demonstrate that the clustering performance can be boosted by exploring the topological structure in data. As expected, the experimental results on a large number of real-world datasets confirm the superior performance of our method. Second, it seems unfair to compare the performance on synthetic dataset with predesigned topological clusters with other traditional methods, considering that our model is designed with the topological structure considered.
>
> R3: To maximize the performance under different datasets, we performed a grid search for the parameters $\alpha$ and $\beta$. We roughly search $\alpha$ and $\beta$ in the range of [0,50]. Better parameter tuning would achieve better clustering results than that reported in this paper.
>
> R4: Real-world datasets are often sampled from a nonlinear low-dimensional manifold which is embedded in the high-dimensional ambient space [1-4], which is well supported by "To compare and classify such observations — in effect, to reason about the world — depends crucially on modeling the nonlinear geometry of these low-dimensional manifolds" in [1], and "Understanding the potential intrinsic low-dimensional structures of those high-dimensional data is an essential pre-processing step for a number of further data analysis processes" in [3]. Hence we argue that it is beneficial to explore the implied data manifold by learning the topological relationship between data points. To do so we propose to explore the implied data manifold and seamlessly integrate multiple affinity graphs into a consensus one with the topological relevance considered. Experimental results on several real-world datasets indicate the effectiveness of the proposed method.
>
> [1] Nonlinear dimensionality reduction by locally linear embedding. Roweis S T, et al., Science, 2000.
>
> [2] On spectral clustering: Analysis and an algorithm. Ng A, et al., NeurIPS, 2001.
>
> [3] Adaptive manifold learning. Zhang Z, et al., IEEE Transactions on Pattern Analysis and Machine Intelligence, 2011.
>
> [4] A unifying framework in vector-valued reproducing kernel hilbert spaces for manifold regularization and co-regularized multi-view learning. Minh H Q, et al., The Journal of Machine Learning Research, 2016.

---

> > ### Comment · Reviewer_wnN6 · 2022-08-09
> > **Thanks for the authors' great efforts during the rebuttal**
> >
> > The responses have addressed my concerns. After reading other reviews and responses from authors, I keep my score as is.

---

> > > ### Author Response · Authors · 2022-08-09
> > > **To Reviewer wnN6**
> > >
> > > We appreciate your positive comments and thanks again for your efforts.

---

### Official Review · Reviewer_2xAj · 2022-07-11

**Rating:** 8
**Confidence:** 4
**Soundness:** 3 good
**Presentation:** 3 good
**Contribution:** 3 good

**Summary:**

This paper proposes a new multi-view subspace clustering method to exploit a common affinity representation using self-expression property. The proposed model explores the implied data manifold by learning the topological relationship between data points. Then, the model integrates multiple affinity graphs into a consensus one with the topological relevance considered. Experimental results on seven benchmark datasets illustrate the effectiveness of the proposed model.

**Questions:**

1) Since manifold learning requires a large amount of computational complexity, so how to handle large data sets with the proposed methods? Are there any speedup tricks?
2) The paper is well written but there are some grammatical errors and expression problems in the paper. It is suggested to conduct further inspections.
3) The detailed information about experimental settings is missing. Is there a significant difference in parameter settings for different datasets? And please provide different parameter settings for all datasets if possible.


**Limitations:**

Yes, the authors have addressed the limitations in Section 5.

**Strengths And Weaknesses:**

Strengths:
+ The theoretical basis is intuitive and sound, and the optimization algorithm is well designed.
+ The convergence of the corresponding algorithm is theoretically and empirically guaranteed.
+ The paper is well written and organized, and the code is available.
+ Extensive experiments are conducted, and the experimental results of the proposed method are promising.

Weaknesses:
- The computation complexity of the proposed model seems to be O(n^3). Hence this model is computationally slow on large data sets.
- The experiments section should be improved. More diverse experimental results, such as ablation analysis, should be available.

---

> ### Author Response · Authors · 2022-08-02
> **Response to Reviewer 2xAj**
>
> Thanks for your valuable comments.
>
> R1: As shown in Alg.~2, there are four steps that mainly determine the complexity of our model. Recall that $n$, $m$ and $c$ denote the number of data points, views, and clusters, respectively, the time complexity for updating $\mathbf{Z}^{(v)}$, $\mathbf{S}$, $c$ eigenvectors of $\mathbf{L}_S$, and view weight $w_v$ are $\mathcal{O}\left(n^2mc\right)$, $\mathcal{O}\left(nm^2c+nmc\right)$, $\mathcal{O}\left(n^2c\right)$, and $\mathcal{O}\left(n^2mc\right)$ respectively. We've also given the time complexity of our algorithm in the Appendix C. Actually in practice, we have $m\ll n$ and $c\ll n$, thus the overall complexity is approximately $\mathcal{O}\left(n^2\right)$, which is acceptable for graph-based methods. Note that this algorithm can be further accelerated without much accuracy loss by adopting several off-the-shell acceleration algorithms, eg., skinny SVD [1] or the Woodbury matrix identity [2].
>
> [1] Fast low-rank subspace segmentation. Zhang et al., IEEE Transactions on Knowledge and Data Engineering 2014.
>
> [2] Accuracy and stability of numerical algorithms. Higham N J., Society for Industrial and Applied Mathematics, 2002.
>
>
> R2: Thanks for the suggestion. We've carefully checked the paper again and corrected some typos.
>
>
> R3: The details of the datasets and implementation have been given in Section 4. The parameters for comparison algorithms are set optimal according to the recommendations in their corresponding paper. The parameter settings of our model are given in Section 4.2. we performed a grid search for the parameters $\alpha$ and $\beta$ in our model. We repeat all algorithms 10 times to eliminate the effect of random factors, and report the average scores. Note that the clustering results of our method are relatively stable for different parameter settings, which demonstrates the robustness of our model. The source codes will also be released at GitHub after the acceptance.

---

> > ### Comment · Reviewer_2xAj · 2022-08-10
> > **Reply to authors' response**
> >
> > Thanks for the response to complexity analysis and parameter searching. The authors have well addressed my concerns. And the authors have fixed some errors in the new version. But I still suggest further revisions for the last version. Overall, this is a good work after revision, I maintain my score.

---

> > > ### Author Response · Authors · 2022-08-10
> > > **To Reviewer 2xAj**
> > >
> > > We appreciate your positive comments. We will continue to revise our paper after the discussion. Thanks again for your great efforts.

---

### Meta-Review · Area_Chair_rXNx · 2022-08-25

**Recommendation:** Accept
**Confidence:** Certain

**Metareview:**

All reviewer agree that this paper is innovative and well-written, so I recommend to accept.

**Award:**

No

---

### Decision · Program_Chairs · 2022-09-14

Accept